# Environmental Influences on the Relation between the 22q11.2 Deletion Syndrome and Mental Health: A Literature Review

**DOI:** 10.3390/genes13112003

**Published:** 2022-11-02

**Authors:** Yelyzaveta Snihirova, David E. J. Linden, Therese van Amelsvoort, Dennis van der Meer

**Affiliations:** 1School for Mental Health and Neuroscience, Faculty of Health, Medicine and Life Sciences, Maastricht University, 6211 LK Maastricht, The Netherlands; 2NORMENT, Division of Mental Health and Addiction, Oslo University Hospital & Institute of Clinical Medicine, University of Oslo, 0315 Oslo, Norway

**Keywords:** 22q11DS, copy number variation, gene-environment interaction, clinical heterogeneity, neuropsychiatric disorders

## Abstract

22q11.2 deletion syndrome (22q11DS) is a clinically heterogeneous genetic syndrome, associated with a wide array of neuropsychiatric symptoms. The clinical presentation is likely to be influenced by environmental factors, yet little is known about this. Here, we review the available research literature on the role of the environment in 22q11DS. We find that within-patient design studies have mainly investigated the role of parental factors, stress, and substance use, reporting significant effects of these factors on the clinical profile. Case-control studies have been less successful, with almost no reports of significant moderating effects of the environment. We go on to hypothesize which specific environmental measures are most likely to interact with the 22q11 deletion, based on the genes in this region and their involvement in molecular pathways. We end by discussing potential reasons for the limited findings so far, including modest sample sizes and limited availability of environmental measures, and make recommendations how to move forward.

## 1. Introduction

22q11.2 deletion syndrome (22q11DS, OMIM #192430/188400) is a genetic syndrome associated with a microdeletion on the long arm of chromosome 22 [1]. The clinical phenotype is highly heterogeneous and may include obesity [2], heart defects, facial anomalies, immune-related issues, and developmental delay [3]. Individuals with this syndrome further have a 20-fold increase in the risk of developing schizophrenia [4,5,6], and it is associated with a range of other neuropsychiatric disorders [7,8] including autism spectrum disorder (ASD) [9], intellectual disability [10], attention deficit hyperactivity disorder (ADHD) [11], Parkinson’s disease [12], anxiety, and depression [13]. Patients with 22q11DS also exhibit a wide range of impairments in the linguistic, affective, and cognitive domains [14,15,16,17,18,19].

While the 22q11.2 deletion strongly impacts mental health, 22q11DS is characterised by a high level of phenotypic heterogeneity [20]. The complex interplay between many genetic and environmental factors that regulate neurodevelopment underlies the significant clinical variability observed among 22q11DS individuals [21]. Identifying the factors explaining this heterogeneity may enable more effective, personalised treatment and more accurate prognoses for 22q11DS patients, and inform our understanding of mechanisms of mental illness more broadly.

The 22q11DS prevalence is 1:3672 based on national registry data [1,22]; there are difficulties for in-depth research due to a lack of adequate sample sizes. However, the aggregation of carriers through large-scale collaborations, such as the Psychiatric Genomics Consortium (PGC) [23] and Enhancing NeuroImaging through Meta-Analysis (ENIGMA) [24], is now making possible well-powered studies with an accumulation of neurobiological information. Other consortia, such as the International 22q11.2 Brain Behaviour (IBBC) [25] and the NIMH-funded Genes to Mental Health (G2MH) initiative [26], in addition to aggregation of data, have enabled deep phenotyping using standardised instruments across research groups. This may allow us to identify those environmental factors that determine to what extent carriers suffer from the detrimental effects of the 22q11 deletions. This is valuable, given that until recently, the focus of 22q11DS research has been concentrating mainly on neurobiology, and environmental factors to date have received little attention.

The goal of this short review is to provide a brief update on our knowledge of the role of environmental factors in moderating the effects of 22q11DS on mental health and cognition and the underlying biological mechanisms. In general, studies looking into the role of the environment in explaining the clinical heterogeneity of 22q11DS individuals have followed two approaches, leveraging either within- or between-group designs (see Table 1).

## 2. Within-Group Studies

One group of studies has investigated environmental effects in cohorts of 22q11DS patients. Studies looking into the role of maternal stress on children’s behaviour problems [28], of substance use on psychosis [35], of the character of a close relationship, of stressful events and neighbourhood danger on anxiety and depression [33], and general history of trauma [39] have reported non-significant findings. Other studies have shown significant effects of the following environmental factors: (1) 22q11DS patients with higher levels of parental anxiety or depression had significant increases in psychopathology [27]; (2) more usage of physical punishment was linked to higher levels of problem behaviours [29]; (3) individuals with higher parental education or parent intelligence showed higher IQ scores [30,31]; (4) higher parental socio-economic status (SES) correlated with higher social competency of patients [32], less frequent behaviour problems [37] or patients’ better global functioning [39]; (5) higher stress load—the number of negative life events, was associated with psychotic symptoms, e.g., hallucinations and delusions, as well as with dysfunctional coping strategies [34]; and (6) higher scores on peer victimisation and hostile close relationships were related to reports of both higher levels of anxiety and depression as well as to impaired tolerance to normal stress in patients [33]. This general overview makes apparent the many gaps in our understanding of the role of the environment in 22q11DS. Most identified studies investigated the role of parental factors, stress, or substance use on cognition or behaviour. More studies looking into other factors of the environment, such as migration [40], poverty [41], patient lifestyle [42], family structure changes [43], social support [44], and drug exposure [45] are necessary, particularly because these factors have generally been implicated in psychosis risk, and 22q11DS is a major known risk factor for psychosis [46] (see Figure 1).

## 3. Between-Group Studies

The other group of studies has employed the gene-environment interaction (GxE) approach, with an interaction term between 22q11DS status and specific environmental factors, in case-control samples. These studies have investigated the moderating effects of activity-related stress on cortisol reactivity [38], parental expressed emotions on children’s behavioural problems [36], SES on cognition [37], and daily-life stressors on affective and psychotic reactivity [18]. However, among these studies, only one had a significant interaction effect. Specifically, higher activity-related stress in 22q11DS was associated with blunted cortisol response [38]. Factors central to within-22q11DS studies, such as parental intelligence or social interactions, have so far received little attention in such case-control studies. One of the possible explanations for this is that the information for parents is often not available for the control group. Detecting differences in the effects of environmental factors between the controls and 22q11DS individuals demands more complex statistical models due to the inclusion of the interaction term [47] requiring bigger samples [48]. Yet, with an increase in the sample size, this approach could provide more insight into underlying mechanisms than the within-group design. Case-control studies have the advantage that differences in response to environmental factors allow us to identify which behavioural domains are most vulnerable. Relative to within-group design, case-control studies allow us to identify differences in sensitivity to environmental factors between 22q11DS and controls, which points towards biological mechanisms underlying the pathology of 22q11DS.

## 4. Biological Mechanisms

A more theory-driven approach to choosing environmental factors for studies would be beneficial, based on the symptoms of 22q11DS, and the possible biological mechanisms involved. For example, biological measures, such as hormones, inflammatory markers, or brain morphology, may mediate the effects of the environment on the clinical heterogeneity of 22q11DS. Neuroimaging studies have identified such mediators—neuroanatomic abnormalities linked to 22q11DS aetiology [49], as well as brain connectivity dysfunction [50]. 

Five genes in the 22q11.2 region [51] have been mainly implicated in the neuropsychiatric phenotype: proline dehydrogenase (*PRODH*) [52], DiGeorge Critical Region 8 (*DGCR8*) [53], catechol-o-methyltransferase (*COMT*) [54], T-box 1 (*TBX1*) [55], and septin 5 (*SEPT5*) [56]. The knowledge of the biological function of these genes and how environmental factors might impair their functionality may further point to potential environmental influence underlying 22q11DS heterogeneity.

*PRODH* codes for proline dehydrogenase, an enzyme that catalyses the first step in proline degradation. It is essential for proline protection against hydrogen peroxide-induced cell death [57]. The presence of an excessive amount of hydrogen peroxide and severe reduction of proline dehydrogenase activity could lead to poor cell response to oxidative stress, and eventually, cell death [57]. Hydrogen peroxide levels are increased by certain environmental stresses, such as smoking, radiation, toxins, and inflammation [58,59], meaning that the reduced expression of *PRODH* combined with environmental stressors makes cells more vulnerable to oxidative stress. Early childhood factors, such as maternal smoking, neighbourhood quality and PRODH activity, could be candidates for future interaction studies.

The DGCR8 protein binds with Drosha—the primary nuclease that executes the initiation step of miRNA processing in the nucleus. The disruption of Drosha function could lead to genome instability and lower its antiviral activity [60]. Stressful events [61] and lower SES [62] are associated with lower pathogen defence, meaning that *DGCR8* hemizygosity—the presence of only one copy of a gene, and related decreased resistance to infection due to environmental factors could contribute to the worsening of the clinical picture in the case of 22q11DS.

The COMT enzyme takes part in the degradation of catecholamines, such as dopamine and norepinephrine, and is connected to a number of mental conditions, such as schizophrenia, ADHD, and bipolar disorder [63,64]. Studies have shown that *COMT* polymorphisms interact with parenting quality to influence attention in children [65], moderate the role of environmental stress on negative affect in nonaffective psychotic disorder patients [66], and moderate the relationships between a number of traumatic events and PTSD risks [67]. Since COMT contributes to various mental disorders, further studies are needed on childhood stress triggers, including low SES, migration, adversity, family structure, and trauma [68].

TBX1 is a DNA-binding protein involved in developmental processes, which has effects on specific cognitive functions (spatial memory and cognitive flexibility) in mice models [69]. Various environmental factors contributing to the same cognitive domains, such as adverse childhood experiences [70], maternal immune activation [71], and stress [72], could enhance the effect of TBX1 malfunction and increase the number of symptoms of neuropsychiatric disorders.

*SEPT5* codes for a nucleotide-binding protein, Septin-5 and its disruption leads to impaired cytokinesis. It is involved in molecular pathways involved in Parkinson’s disease [73]. Environmental influences that might contribute to Parkinson’s disease, such as toxins (neurotoxin MPTP [74], pesticides, solvents, metals [75]), and infection [76], could also be investigated as potential interaction factors with 22q11.2 deletion. Closer attention to stress factors, maternal infection, toxins, and substance use during pregnancy of a mother and the first years of a patient’s life in research might lead us to a better understanding of clinical trajectories of 22q11DS.

## 5. Future Directions

The lack of research into the moderating role of environmental factors can for an important part be attributed to the limited environmental data available for 22q11DS. Differences in data collection and instruments between the clinical cohorts with 22q11DS patients’ information further hinder the identification of interaction effects, as they impede aggregation of the required sample sizes. One of the solutions could be standardised and comprehensive questionnaires among research teams working on 22q11DS covering all likely categories of environmental factors affecting carriers, for example, maternal stress, birth complications and infections, SES (parent’s education, urbanicity, migration, poverty), lifestyle (substances use, sleep quality, physical activity), home environment (parenting, family environment), social environment (harassment, trauma, stressful events), and cognitive tests evaluating different cognitive domains, as implemented by the G2MH consortium [26]. Between-cohort differences in recruitment further add to difficulties in identifying the causes of clinical heterogeneity, as population cohorts tend to contain relatively less-affected cases than clinical cohorts, hindering cross-cohort analyses.

Pending data availability, research into the role of ethnicities, such as analyses of non-European and non-American samples [77,78], may contribute to a greater understanding of the clinical heterogeneity of the disorders associated with 22q11DS. Furthermore, transparency in data sharing should be reached, ideally through the creation of a common database with widely available access for researchers. The different age ranges in the mentioned studies make it challenging to generalize the results, necessitating separate analyses for children and adults. As well as the data for children and adults, data for similar patients across the lifespan will provide researchers with information about how 22q11DS representation changes with age. Additionally, the collection of data from parents and siblings of 22q11.2 deletion carriers is valuable to evaluate the environmental influence of the family-determined factors, as in the G2MH consortium.

Research has further shown that besides the influence of environmental measures on the presentation of the 22q11DS, the heterogeneity of the symptoms of 22q11.2 deletions carriers highly depends on (1) the size of the deletion [49], (2) the particular part of the 22q11.2 locus that is deleted [79], (3) the developmental stage in which CNVs occur (leading to mosaicism) [80], and (4) other genetic variables, such as CNVs in other regions and SNPs in associated genes [81,82]. Hence gene × gene interactions are potential research targets, along with the above-mentioned G × E analyses. There is a genetic component in certain environmental measures, such as parental education or SES. Clarification of the role of the environment in 22q11DS requires us to tease apart this genetic component from the environmental influence.

## 6. Conclusions

It is evident that the role of the environment in explaining 22q11DS clinical heterogeneity has remained under investigated. With an increase in the number of recent papers on 22q11DS-disorder and 22q11DS-environmental factors links within the affected individuals’ group, a lot of progress is expected in upcoming years, through studies involving interaction between 22q11DS status and environmental factors. Integration of knowledge from different fields, such as genetics, biochemistry, and cognitive neuroscience, may be leveraged to identify several potential targets for interaction analysis in 22q11DS research. Using the knowledge of the genes involved in the pathology of 22q11DS, their function and environmental factors contributing to their activity could lead to biology-based theories of gene × environment interaction in 22q11DS, inspiring interaction models that explain clinical heterogeneity. Ultimately, knowledge of environmental factors involved in 22q11DS and their influence on the biological level could lead to more personalised medicine and better clinical outcomes in the future.

## Figures and Tables

**Figure 1 genes-13-02003-f001:**
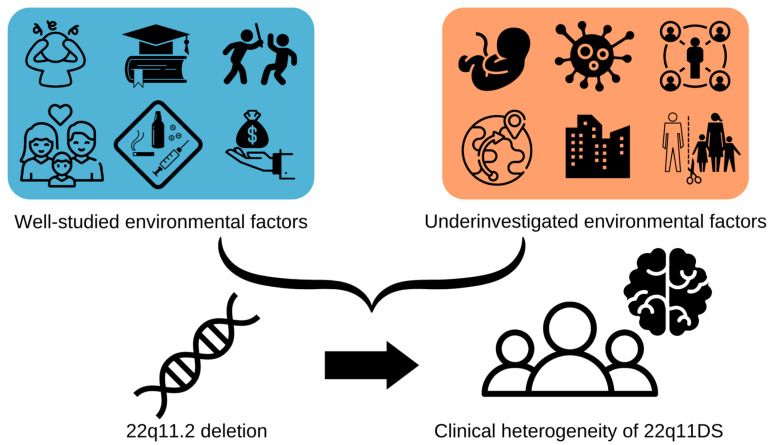
Visualisation of types of environmental factors, with on the left side those studied in relation to 22q11DS clinical heterogeneity (stress, parents’ education, peer violence, parents’ psychopathology, substance use, socio-economic status) and on the right side those that remain under investigated (birth complications, infection, social interactions, migration, urbanicity, family structure).

**Table 1 genes-13-02003-t001:** Overview of the reviewed sources grouped by the type of the study separately for within 22q11DS and between-group studies.

Author	Age Range	Sample Size	Environmental Factor	Outcome Measure	Significance *
Within 22q11DS					
Parental influence					
Sandini et al., 2020 [27]	Patients > 11 years old	103 patients in a total sample	Parental anxiety and depression	Children’s psychopathology	+
Briegel and Andritschky 2021 [28]	4–14 years old and their mothers	41 children for the analysis	Maternal stress	Children’s behaviour problems	-
Allen et al., 2014 [29]	9–18 years old	48 children and adolescents	Parenting style	Children’s behaviour problems	+
Klaassen et al., 2016 [30]	5.2–15.9 years old	171 children	Parental Intelligence	The intelligence of the offspring	+
Socio-economic status (SES)					
Olszewski et al., 2014 [31]	11.8 ± 2.0 years old	73 children	Parental education	The intelligence of the offspring	+
Shashi et al., 2012 [32]	10.5 ± 2.6 years old	66 children	Parental SES	Social competency	+
Stress					
Gur et al., 2021 [33]	24.6 ± 9.3 years old	80 patients	Peer victimisation	Reports of anxiety/depression	+
Armando et al., 2018 [34]	12 to 25 years old	59 patients	Past stress load	Dysfunctional coping strategies	+
Substances					
Vingerhoets et al., 2019 [35]	30.91 ± 12.65 years old	434 adults	Substance use	Psychosis	-
Between-groups					
Parental influence					
Serur et al., 2022 [36]	3–8 years old	24 children with 22q11DS, 28 children with idiopathic ASD and 23 typically developed children	Parental expressed emotions	Children’s behavioural problems	-
Socio-economic status (SES)					
Shashi et al., 2010 [37]	10.2 ± 2.6 years old for patients	65 children with 22q11DS and 52 controls	SES	Cognition	-
Stress					
van Duin et al., 2019 [38]	34.11 ± 9.81 years old for patients	27 adults with 22q11DS and 24 age/sex-matched healthy controls	Activity-related stress	Cortisol reactivity	+
Schneider et al., 2020 [18]	34.11 ± 9.81 years old for patients	27 adults with 22q11DS and 24 matched controls	Daily-life stressors	Affective and psychotic reactivity	-

* Indication of whether the results were significant (“+”) or not (“-”) was based on the study-defined significance threshold.

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
