# Peer review of "Environmental Influences on the Relation between the 22q11.2 Deletion Syndrome and Mental Health: A Literature Review"

_genes, 2022, doi:10.3390/genes13112003_

Round 1

Reviewer 1 Report

Comments to the Authors:

The authors review the environmental factors that affect neurodevelopment in 22q11.2 deletion syndrome. It also highlights the under-investigated environmental factors effects on mental health of individuals with 22q11.2 DS as well as gene-environment interaction. The article is very well-written, and it adds to the general knowledge about 22q11.2 deletion syndrome in general and environmental influences on the mental health of individuals with 22q11.2DS.

There are very few points to be considered which would be beneficial for this paper.

Page 2, Line 60: “higher stress load – a number of negative life events, was associated with positive and general psychotic symptom domains as well as with dysfunc- 60 tional coping strategies. Would you please clarify “positive” here?

Page 2, Line 73: Define “SES” and in Figure 1 please.

Page 3, Line 82: Needs to cite references.

Page 3, Line 103-108: needs to cite references

Table 1, page 6: van Dyin et l. 2019. In the third column: “27 adults and 24 age/sex-matched healthy controls”. I believe it 27 adults with 22q11DS. You might need to specify this.

Reviewer 2 Report

This is an interesting article but please add the OMIM number of the syndrome at the beginning of the introduction

Please clarify in the title that it is a review article

Reviewer 3 Report

In this short review, Snihirova and colleagues make an overview of published studies aimed at evaluating the contribution of environmental factors in the heterogeneous neuropsychiatric and neuropsychological clinical manifestations associated with 22q11.2 deletion syndrome (22q11.2DS).

The topic is interesting. The article has not a clear structure, resulting slightly difficult to follow. However, English is adequate and the principal considerations regarding studies, methodologies, implications and limits are well synthesized.

Please, finds below some minor comments:

  Evaluate the possibilities to subdivide the article in small sections (e.g. Introduction; Within group studies; Case-control studies; Genes-Environment interaction; Conclusions), to make the article more readable.

-      Since cited articles display distinct age ranges for the evaluation of subjects, please discuss briefly if some implications or limitations are known about this aspect.

-      A legend for Table 1 should be included in the manuscript, particularly for “significance” +/-.
